# Consumer Trust in Quality and Safety of Food Products in Western Siberia

**Djamilia F. Skripnuk** [1] , **Vladimir A. Davydenko** [2] , **Gulnara F. Romashkina** [3] and **Roman R. Khuziakhmetov** [4,*]

1   Institute of Industrial Management, Economics and Trade, Peter the Great St. Petersburg Polytechnic University, St. Petersburg 195251, Russia; djamilyas@mail.ru
2   Institute of Finance and Economics Research Center, Institute of Finance and Economics, University of Tyumen, Tyumen 625003, Russia; vlad_davidenko@mail.ru
3   Department of Economic Security, System Analysis, and Control, Institute of Finance and Economics, University of Tyumen, Tyumen 625003, Russia; gr136@mail.ru
4   Department of General and Economic Sociology, Institute of Finance and Economics, University of Tyumen, Tyumen 625003, Russia
*   Correspondence: r_o_m_a_n_14@mail.ru

**Abstract:** Modern Russia faces difficulties in ensuring food quality and safety. The updated federal food security doctrine focuses on export opportunities and monitoring the ratio between domestic production and consumption. This agenda is determined by possible external threats: sanctions and trade wars, various conflicts, and economic and agricultural crises. The aim is to reveal the features of consumer behavior when interacting with food operators and to show the influence of socio-economic characteristics on individual practices. Empirical data are obtained from the authors' mass survey and in-depth interviews. Results might signify that society came to a consensus on trust in quality and safety of food. Observed differences in outlet and product choices can be explained by income, settlement type, and age. Local producers are struggling to enter retail chains, as there are contradictions between consumer expectations and internal policies of sales operators. Experts argue that people poorly assess the real risks of economizing and favoring low-quality food.

**Keywords:** trust; food security; food quality; food safety; retail chains; local producers

## 1. Introduction

New external challenges, i.e., globalization of food trade, growing world population, climate change, and internal challenges, i.e., rapid modification of food system, archaization of technologies, domestic political obstacles, and increased relevance of food security issues for countries and regions.

Whereas safe food supply stimulates sustainable development, supports national economies, trade and tourism, unsafe food creates a vicious cycle of disease and malnutrition, affecting various population groups. International organizations aim to enhance the capacity to prevent, detect, and respond to threats associated with poor quality diet [1].

Economics and formal modeling tend to consider consumer as a result of global food policy, not as its central element. However, a change in consumer behavior is a necessary condition for the food security problem solving [2]. Thus, food safety is supported by the actions of agricultural producers, food sellers and consumers.

The authors implement the FAO's food security/insecurity framework, which addresses various aspects of consuming adequate amounts of quality food, as well as its physical accessibility and economic affordability [3]. Methodological consideration of a scale of risk insecurity allows us to regard food security as the absence of food insecurity.

The interpretation of food security has been changing historically, politically and economically. Traditionally, most food-related strategies have rarely viewed nutrition as

their primary goal or primary concern [4–6]. However, the concepts of food security and food safety are interrelated and partially overlap [7].

People with moderate food insecurity, suffering from a limited ability to obtain food, encounter uncertainty and are forced to compromise on quality and/or quantity of food they eat (this is especially true for families with children). It should also be noted that most moderately food insecure countries are highly dependent on international commodity trade, including Russia [2].

In today's Russia, the problem of food affordability remains important. It is necessary to understand what happens when real incomes decline: whether people will economize on food and to what extent. In times of crisis, retail trade turnover is reduced as well as the choice of quality products, while nutritional balance is disturbed [8]. Moreover, a change in modern people lifestyle provokes new health threats, which are caused not only by malnutrition or a meager food basket, but also by diet imbalance and low-quality product purchases.

Despite the cooperation of Russian government with FAO, the essence of the food security doctrine developed in Russia in 2008 is understood differently. While basic principles—the need for accessibility and affordability of food at any time in any place, as well as safety and quality of food–are fully consistent with the FAO concept, target indicators, used in official documents, allow for thinking that authorities are mainly focused on import substitution. As budgetary allocations are directed to renew agricultural machinery and equipment (paying special attention to domestically manufactured), and to support businesses, which can provide analogues to the food previously purchased abroad [9,10].

Principal indicators of the doctrine, planned for the period of 2010–2020, were achieved by 2018. Such rapid success in food autonomy provision can be attributed to the policy of sanctions and anti-sanctions, in particular, to presidential decree banning/restricting import of agricultural products. As a result, in 2018, Russia achieved more than 90% self-sufficiency in meat products, 84% in milk, and 62% in fruits. Deliveries of European products to Russia fell almost by half from 2013 to 2018. However, as evidence shows, Russian consumers "lost" from such a policy, as the share of food spending in their household expenditures increased from 40 to 60% for different product groups. In addition, the share of counterfeit dairy products and shadow turnover of fruits and berries increased [11].

The updated food security doctrine in Russia uses a new assessment system. From discussing import substitution, officials moved on to considering export opportunities and monitoring the ratio of domestically produced products to those de facto consumed by Russians. This formulation of the question is still determined by the country's security in the face of possible external threats: sanctions and trade wars, various conflicts, economic and agricultural crises [12].

This comprehensive program was adopted by the state to provide integrated development of rural areas through infrastructure projects, labor resources advancement, import substitution and export orientation. Nevertheless, the document still has a long way to go to recognize and uncover the deep problems that have been brewing in Russian agriculture over the past 30 years.

Russia faces difficulties in ensuring food security, quality and safety. For example, Barsukova points out that, even in the most stagnant periods of the Soviet time the consumption of such important food products as milk was at the height unattainable today [13]. Analyzing the Russian agrarian industry, Kalugina notes that control over food production safety is almost impossible in the process of the gradual archaization of agriculture. The diet and nutrition structure deteriorated sharply in the 1990s. Even now, the share of low-quality products (including imported ones) is high, which leads to a significant deterioration in population health. High profitability of low-quality and counterfeit foods leads to unprofitability of quality food production. As a result, people often risk their health buying cheaper products. For instance, dairy products are mainly bought by wealthier

citizens, while low-income citizens, trying to compensate, consume increased amounts of potatoes and baked goods" [14].

Shagaida and Uzun claim that full satisfaction of population needs for meat and meat products is achieved by increasing consumption of cheaper and less nutritious (or even harmful) food: for example, cheap meat processing products, etc. [15]. A thorough analysis of official reports on assessment of food security in Russia challenges the statement that complete food security is achieved, and this is especially true for the regions outside capitals [16]. Using the empirical data, it is intended to prove these assumptions below.

As noted by Bilali, despite the high support for an integrated consideration of food security and nutrition, this approach is still rare in academic publications [7]. It is even less common in publications based on Russian material. Hence, the current work is expected to fill the gap of research in this area, which, to some extent, ensures the significance of results and conclusions of this work.

The paper aims to reveal the features of consumer behavior when interacting with food operators and to show the influence of socio-economic characteristics on individual practices. In the framework of ensuring food security, the problem of public trust in quality and safety of food will be considered as an attempt to reveal how and where people buy food. New practices of food distribution are widely developing now: online orders, hyper- and mega-markets, etc. This research poses a question of whether there have been qualitative changes in food consumers' behavior, namely, to what extent their practices are diversified depending on a type of the settlement, level of income and social status. In order to specify the problem, the following food retail operators are examined: (1) big stores (super-, hypermarkets), retail chains; (2) markets, fairs; (3) convenience stores; (4) firm stores "from the producer".

Having studied a wide range of academic works, the authors formulate nine research hypotheses (H1–H9):

**Hypotheses 1 (H1).** *Most of the respondents trust in quality and safety of food.*

**Hypotheses 2 (H2).** *Respondents' trust in quality and safety of food is higher than in government, governor, local authorities and business community.*

**Hypotheses 3 (H3).** *Firm stores (from the producer) hold a leadership position in terms of trust among sales operators.*

**Hypotheses 4 (H4).** *A significant part of behavioral characteristics of food consumers are determined by such factors as gender, income, social status, education and settlement type.*

**H5**–**H8** deepen **H4**. It is assumed that wealthier and educated people tend to make a more conscious choice, which strengthens their trust in quality and the safety of purchased food. These assumptions provided the foundation for **H5** and **H6**. Furthermore, it is expected that wealthier and educated urban populations possess higher trusts in long food supply chains (chain stores and supermarkets) compared to short ones (food bases, markets, fairs, firm stores "from the producer"). These assumptions provided the foundation for **H7** and **H8**:

**Hypotheses 5 (H5).** *The level of trust in food quality and safety for all sales operators is increasing in higher-income groups.*

**Hypotheses 6 (H6).** *The higher education level people have, the higher the person's trust in quality and safety of food.*

**Hypotheses 7 (H7).** *Urban dwellers trust big stores and retail chains, whereas rural dwellers prefer convenience stores, markets, and fairs.*

**Hypotheses 8 (H8).** *The younger the respondents are, the higher their trust in firm stores (brand choice) and retail chains (sales regulation choice).*

**Hypotheses 9 (H9).** *There are significant conflicts between the interests of sales operators and consumers.*

Scientific novelty of the study lies in the offer to shift the focus of research on food security, rethinking the consumer role. It may be insightful to address an individual not as a final link in a supply chain but as its central element. The authors propose to consider a consumer not as the result of a long succession of interactions between various participants in the food market, but as a determining initial factor of possible and necessary changes.

## 2. Background

### 2.1. Individual Perception of Food Quality

Modern consumers tend to be more health-conscious. When buying food, they are increasingly likely to expect and demand more nutritious and fresh products. As a result, "clean labels", that claim the product to be organic, fresh and chemical-free, have gradually gained attention among consumers [17].

Another matter of consumer concern is the apparent lack of control and safety of agri-food production. This explains substantial changes in European food policy and consumer behavior. A regulatory framework, based on risk assessment, control, management and communication, aims to improve food safety and health standards. Such an integrated approach fosters the growth of agricultural businesses in the developed countries through the provision of higher safety guarantees. At the same time, some companies consider today's consumer demand as an opportunity to gain their competitive advantage by presenting more credible information about a product's quality [18].

Food quality reflects the level of consumer satisfaction, depending on whether a product's characteristics meet personal requirements. Thus, quality has both objective and subjective dimensions: the first one is associated with chemical constitution and nutritional value of a product, and the second manifests the needs and preferences of an individual consumer. Therefore, food quality, as a complex construct, builds upon a diversified set of attributes, which can be divided into three groups [19]. It is rather easy to distinguish between the search and experience attributes. The first group implies the idea of a pre-purchase assessment (size, color, and price), while the second one refers only to the post-purchase stage (taste and flavor). The third set of attributes—credence—cannot be ascertained either before or after purchase, and requires supplementary information.

To be more precise, as Migliore et al. claim, perception of quality revolves around personal needs: food safety and production process, environmental impact, ethical content (fair wages, animal welfare), and production area [20].

According to the idea that consumer perception influences the way food quality is interpreted, Petrescu et al. argue that the key to understanding how food quality evaluation functions is in detecting the cues used by consumers. The authors believe this to be a highly relevant matter because it forms the ground for the consumer's desire to buy [21]. It can be concluded that differences in food quality perception are common over time and places; and this complex and dynamic phenomenon requires constant examination and update.

Quality evaluation consists of two stages: pre-purchase and post-purchase. At the point of purchase, consumers use both explicit (for example, price and claims) and subtle cues (for example, packaging graphic design, material) [22].

Product descriptions have proved to be a solid determinant of consumers' perception, encouraging markets to grant reliable data in order to be functional. Therefore, labelling, as a means of providing information on the product characteristics related to environment and health, is a widespread instrument allowing consumers to make a well-founded choice. Studies largely report that the provision of detailed, credible, and transparent signals through packaging contributes to establishing a profound information-based interaction and positively affects consumers' behavior by increasing trust [23].

The authors' empirical data also address the issue of individual perception of food quality and aim to reveal substantial factors that the regional population pays attention to when choosing food product groups in different outlets.

## 2.2. Trust in Food Products

Food producers and sellers use explicit and intentional communication with clients. The first type focuses on potential customers and has higher costs because it requires the allocation of extra resources aimed at establishing more personalized relationships with clients. The second one is expected to maintain positive relations with existing customers by using impersonal and mass-marketing tools. Dana et al. states that, though largely different, both communication types have some essential features in common, such as corporate heritage, which represents a patrimony of history deriving from the accumulation of reputation over time, trust, values and prestige, handed down through the generations, as well as corporate ethical values, which improve loyalty [24].

The concept of trust can be viewed from multiple angles and its understanding differs significantly across academic fields [25]. According to the examination of famous sociological discussions, it is possible to reveal the following theoretical nuances: Coleman developed a rational-reflexive version, Giddens referred to trust primarily in terms of "double structuring", Goffman emphasized socially integrative function of action, Luhmann addressed perspectives of managing social complexity. In the context of ongoing globalization, neither social trust nor the norms of reciprocity are still universal [26].

Many thoughts and ideas on trust can be found across a large range of disciplines. The authors adopt a conceptualization of trust from Lobb's work, who suggests considering it as a combination of rational thinking (cognitive process) and feelings (affective influence), which often depend on previous experience. The first component of this integrated definition approaches "trust" as "a functional alternative to rational prediction aimed at reducing complexity", while the second one sees "trust" as an "extent to which one believes that others will not act to exploit one's vulnerabilities" [27,28].

It is important to consider the concept of trust not only in terms of risks and food safety, but also from the institutional standpoint, regarding suppliers and regulators. The efficiency of food safety projects relies heavily on purchasing behavior, which in its turn, gets greatly affected by who, how and why transfers certain information. To some extent, consumer's trust in an "institution" or an individual seller they purchase from, must be unconditional as consumers are fully reliant on a provider's reputation and a regulator's competence [29].

Trust in products and services has long been in the center of scientific research. Commitment-trust theory emphasizes that trust is a prerequisite for maintaining long-term relationships with the company. Brand trust, for its part, is viewed as "average consumer willingness to rely on the ability of a brand to perform its stated function". It is also reported that brand image contributes to trust [30].

Consumer's trust differs between people in developing and industrial countries. The latter tend to place greater trust in public institutions and less in suppliers, which is not surprising at all, due to the fact that the rule of law is often executed more credibly in the industrial countries. For instance, Le et al. states that it is often more difficult to control the market in the developing countries because supply chains are mostly non-state and consist of disparate households [31].

The results of the current study, addressing the Russian case, are expected to be more comparable to the example of the developed countries.

## 2.3. Safety of Food Products

Food safety is one of the main concerns for people and health agencies, as the vast majority of population contacts a foodborne disease at least once in a lifetime. It is impossible to underestimate the importance of clean food consumption. Since food can be contaminated or harmed when moving through any supply chain stage, each party is responsible for ensuring that one's meal will not cause damage. Considering the fact that society needs to prevent this from happening, scholars and administrators strive to gain a deeper understanding of factors that cause quality problems at any point along supply chain, including production, distribution, storage, cooking, and consumption [32].

As Ortega and Tschirley note, food safety is uneven around the world. Even though foodborne illnesses occur in the developed countries, they affect a small number of consumers. While in the developing countries, such issues are considerably more wide-spread due to a lower level of economic development, which is associated, among other things, with poor sanitation and water quality, fraud, adulteration, toxic residues from pesticide and additives, and information asymmetries resulting from opportunistic behavior [33].

Food safety standards are governmental tools to cope with food safety issues in the supply chain. These standards are supposed to ensure that food safety is not a matter of choice for consumers. However, due to the transformation stage they are facing, the emerging markets may struggle to endeavor a high baseline of food safety because the entire industry is not ready to upgrade resources. At the same time, incomes are becoming higher, thus making consumers be more interested in food safety and quality, rather than just quantity, which predetermines the necessity to improve control over technology, production and additives in domestic market. Therefore, governments of the developing countries are gradually updating food safety standards in the attempt to distribute the cost of technological and processing upgrade on many actors. This process also contributes to minimizing difficulties for poorer individuals who cannot easily adapt to an increase in food prices, which is inevitable when safety and quality standards improve [34].

In recent years, some international organizations (WHO, FAO) have developed programs and guidelines to prevent or control conditions leading to contamination along the farm-to-fork continuum. Evidence shows that the implementation of on-farm food safety (OFFS) practices can prevent, or at least reduce, contamination in various products. Nevertheless, producers are uncertain about the effectiveness of these programs and hesitate to adopt new standards. Thus, while some practices are perceived to be excessive; others are believed to be inadequately executed. As Rezaeri et al. claim, high contamination levels are assumed to be caused by the lack of attention to management practices and public health [35].

It is assumed that the regional production in Russia may experience the food safety problems that are typical for the emerging markets, including irregular machinery cleaning, negligent quality testing of raw materials, lack of required equipment, and insufficient observation of health criteria.

### 2.4. Local Producers and Food Supply Chains

Successful interaction between stakeholders within the food supply chain is very important. Time, cost, flexibility, and quality are crucial for the supply chain performance, while its management stands upon long-term partnership and investment in technology. Cooperation within supply chain has a positive impact on product quality, cost reduction, and responsiveness [36].

Maintaining high food quality is believed to be the most important indicator for the performance assessment in the supply chain, which can be carried out according to the following five dimensions: trust, communication, flexibility, cooperation, and atmosphere. Ding et al. stated that four elements in the supply chain management are positively related to food quality and safety, including trust, strategic alliance, commitment, and information quality [37].

Support programs, implemented by the Russian regional authorities, predominantly intend to help local producers promote their food. This article pays special attention to the difficulties that local farmers and enterprises face while attempting to enter the retail. This view is determined by the integrated territorial paradigm, which aims to reinforce the capacity of agri-food systems to valorize specific territorial resources and social relations between neighbor-regions. Lamine et al. claim that, to a large extent, territorial embeddedness is a common denominator for the emerging practices of the alternative, local food geography. Agri-food systems are territory-specific; they are integrated into nature and landscape conservation, tourism, and education [38].

Local food systems (LFS) can be defined as a more sustainable alternative to the globalized ones. The supply chain processes, including food production, processing, distribution, and consumption, are embedded in the LFS, which contributes to the evolution of communities and regions, and assures their self-reliance. LFS are introduced in specific regions to make them more economically viable for farmers and consumers, and encourage ecological production and distribution practices. Economic development of LFS and their growth depends on local producers, food industries, distributors and retailers. Locally produced foods are more likely to be consumed fresh. In addition to that, LFS requires no shipping or packaging thus, the carbon footprint is reduced [39].

Systems of local food production and distribution are getting great attention today. This happens largely because the consumer's motivation to purchase locally produced food is becoming more evident. Globalization of food production increased complexity of food supply chains and brought some negative consequences for the environment. However, in these circumstances, local food producers cannot be competitive enough, due to the relatively high logistics costs. Hence, short food supply chains require immediate solutions in terms of distribution and logistics, with an ultimate aim to ensure sustainability and improve competitiveness of local farmers and enterprises [40].

Short food supply chains (SFSC) are originally defined as an example of the farmer's "resistance" to modernize their production and distribution, under the pressure of global retail chains. The resistance is reflected in the fact that direct sales bypass intermediaries, thereby boosting profits for producers, providing better visibility and identifying new niche markets. Based on wide literature review, Dana et al. derive four prevalent characteristics of SFSC:

- geographical proximity—eographical area where food is produced and/or distributed;
- economic viability of the actors involved, mainly from the primary producers' point of view;
- social interaction—producer—consumer interconnection and interaction within communities;
- environmental sustainability [24].

## 3. Materials and Methods

The sociocultural approach shapes the general methodological framework of this article: the region is defined as a territorial community, formed by the activities of social actors (such as residents, groups, organizations). This community performs certain functions in relation to its actors and motivates them, in particular, and the entire society, in general.

The study of the social and economic well-being of the regional population includes the analysis of typological characteristics, living standards, main problems, and challenges people encounter, as well as their trust in state and public institutions.

The article also investigates the consumer's behavior related to food consumption, and suggests the mechanisms to improve the level of food security in the Russian regions.

Empirical data, both quantitative and qualitative, are obtained from sociological research conducted by sociologists of the University of Tyumen (with authors' participation) in April–May 2018 in the Tyumen Oblast. This is a Western Siberian territory, often referred to as "the backwoods of Russia", with the population characterized as neither too poor, nor too rich. The study uses 51 interviews with the owners of agricultural enterprises (23), experts in food retail trade (9), state and municipal authorities (9), and scholars investigating related issues (10).

The mass survey was conducted among residents of the Tyumen Oblast ($n = 1610$). The sample is multistage, representing sex, age and settlement type (urban, rural). Sample structure: 66%–urban, 34%–rural residents; 44%–men, 56%–women. The confidence level is 95%, and the sampling error for one indicator is $\Delta < 3\%$.

The questionnaire included the following two questions.

Q1. Please evaluate the level of trust in quality and safety of food products that you buy: in large stores (hyper- and supermarkets) and retail chains; in the marketplace and at

fairs; in convenience stores; firm stores. Answer options: I completely trust (1); I largely trust (0.75); Not sure (0.5); I don't really trust (0.25); I don't trust at all (0).

Q2. Please indicate where you and your family members usually buy food? If you buy food from local producers, where do you do it most often? (several options are possible).

The authors' analysis also considered the type of settlement (rural/urban), the respondent's age and education, as well as their self-assessment of income and social status (industry, occupation, position and number of subordinates).

Listed below are the answers to the question "Which of the following statements best describe your/your family's current financial situation?". Answers are given in six groups in accordance with the classification regularly used in the previous works of the authors:

- "the poorest"—not enough money for everyday expenses;
- "the poor" —the entire salary is spent on daily expenses;
- "the unsecured" —enough money for everyday expenses, but buying clothes is difficult;
- "the secured" —mostly enough money, but a loan is required to purchase expensive items;
- "the wealthy" —enough money to afford almost everything, but loan is required to purchase real estate;
- "the rich" —everything can be afforded.

Appendix A shows a fragment of the expert interview guide, included in the analysis.

Hypotheses are verified based on unique empirical data. **H1**–**H8** are verified quantitatively. In particular, **H4** is verified using the Kruskal-Wallis test and median test, since the scale for measuring the level of trust in quality and safety of food, as well as the level of trust in sales operators is ordinal. The index method is used to measure the average level of trust. **H9** is verified qualitatively by analyzing narrative data obtained from the expert interviews.

## 4. Results

In terms of trust in food quality and safety, firm stores (from the producer) rank first with the trust index (TI) of 0.68 for all social groups (Table 1). In addition, 62% of the region's residents show their trust in firm stores. Big stores and retail chains rank second (TI = 0.59, trusted by 51%) and are followed by marketplaces (TI = 0.59, trusted by 51%) and convenience stores (TI = 0.57, trusted by 43%), respectively.

**Table 1.** Level of trust in quality and safety of food people buy in different outlets. Answer question 1–Q1 (see Materials and Methods section).

| | % of Respondents Who Trust (Fully or Partially) | Trust Index * | Standard Derivation |
|---|---|---|---|
| Quality and safety of food products that individuals buy: | | | |
| Firm stores "from the producer" | 62% | 0.68 | 0.24 |
| Big stores, retail chains | 51% | 0.59 | 0.26 |
| Marketplaces and fairs | 45% | 0.59 | 0.24 |
| Convenience stores | 43% | 0.57 | 0.26 |
| Authorities and public institutions: | | | |
| Governor | 42% | 0.55 | 0.56 |
| Regional government | 41% | 0.53 | 0.55 |
| Entrepreneurs, business community | 29% | 0.45 | 0.48 |

* Calculated as a weighted average so that "1" corresponds to "everyone trusts completely" and "0" corresponds to "everyone has no trust at all". Source: authors' research.

The index of trust in sales operators in quality and safety of food products turns out to be higher than the index of trust in all regional government institutions (by a small

margin). Evaluation of trust in the governor (the traditional leader of trust among regional institutions) [41], the regional government, and the business community are compared in Table 1 (with a larger margin, and lower overall estimates). This might signify that society shows unanimity in terms of trust in quality and safety of food.

It should be noted that the richer the respondents are, the higher their trust in the food they buy. (Figure 1). This correlation applies to all sales operators. Differences occur in the middle of the ranking. Respondents who have sufficient income to afford shopping ("the rich", "the wealthy", "the secured") put big stores and retail chains second. Representatives of the lowest groups ("the unsecured", "the poor", "the poorest") do not have an opportunity to choose the outlet.

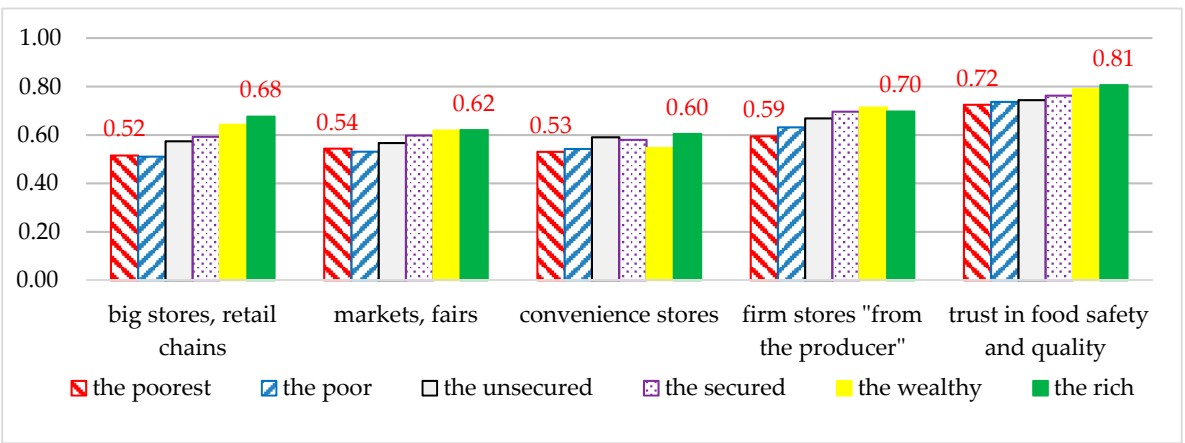

**Figure 1.** Index of trust in regional sales operators in terms of food quality and safety according to the respondent's income. Source: Authors' research.

The level of trust in brand stores is the highest among all social groups, while the ranking of other sales operators varies from one target group to another. The Kruskal–Wallis test and median test are used to check the significance level of the analyzed correlations (Table 2). The income remains the main differentiating factor for all parameters measured in this paper. Variations in the level of trust in food quality and safety are statistically stable, and are explained by the parameters of income and age. In case of identifying features (gender, social status, type of settlement, education), H0 is put forward, which highlights the absence of statistically significant differences.

Consumer behavior when choosing outlets largely depends on income, gender, age, and settlement type. The attitude to markets and fairs depends only on income and gender. Social status and education turn out to be the least significant out of all differentiating factors considered.

Young people trust in food quality and safety more (TI = 0.66) than respondents over the age of 65 (TI = 0.55). However, chain stores rank second when subgroups are distinguished.

On average, residents of rural settlements trust in sales operators less, but the differences are statistically stable not for all outlets. Women are less inclined to trust sales operators (Figure 2), although gender differences in relation to brand stores are insignificant (as H0 is accepted), i.e., men and women behave similarly. Gender differences are more noticeable among rural residents, while, in cities, these distinctions are almost completely erased. Convenience stores are particularly important for the older respondents (contrast is small, but statistically significant). The younger the respondents are, the more they trust they have in firm stores and retail chains; statistical significance of this feature is confirmed by the Kruskal–Wallis test and the median test (Table 2).

**Table 2.** Nonparametric tests (Kruskal–Wallis) verifying differences in the level of trust in food quality and safety for different subsamples.

| Grouping Variable | | Trust in Food Safety and Quality | Big Stores, Retail Chains | Markets, Fairs | Convenience Stores | "From the Producer" Firm Stores |
|---|---|---|---|---|---|---|
| Financial status | Chi-square | 62.45 | 84.04 | 39.92 | 13.44 | 23.38 |
| | d.f. | 3 | 3 | 3 | 3 | 3 |
| | Sign. | 0.000 | 0.000 | 0.000 | 0.004 | 0.000 |
| | (*) | **H1** | **H1** | **H1** | **H1** | **H1** |
| Gender | Chi-square | 1.25 | 13.27 | 4.37 | 4.21 | 1.45 |
| | d.f. | 1 | 1 | 1 | 1 | 1 |
| | Sign. | 0.263 | 0.000 | 0.036 | 0.040 | 0.229 |
| | (*) | H0 | **H1** | **H1** | **H1** | H0 |
| Type of settlement | Chi-square | 0.65 | 15.99 | 4.76 | 28.71 | 10.78 |
| | d.f. | 2 | 2 | 2 | 2 | 2 |
| | Sign. | 0.722 | 0.000 | 0.092 | 0.000 | 0.005 |
| | (*) | H0 | **H1** | H0 | **H1** | **H1** |
| Age | Chi-square | 61.13 | 23.82 | 3.20 | 8.65 | 14.83 |
| | d.f. | 5 | 5 | 5 | 5 | 5 |
| | Sign. | 0.000 | 0.000 | 0.670 | 0.124 | 0.011 |
| | (*) | **H1** | **H1** | H0 | H0 | **H1** |
| Social status | Chi-square | 0.27 | 13.89 | 0.54 | 2.55 | 3.09 |
| | d.f. | 2 | 2 | 2 | 2 | 2 |
| | Sign. | 0.875 | 0.001 | 0.762 | 0.280 | 0.213 |
| | (*) | H0 | **H1** | H0 | H0 | H0 |
| Education | Chi-square | 3.25 | 7.46 | 2.00 | 5.87 | 2.51 |
| | d.f. | 2 | 2 | 2 | 2 | 2 |
| | Sign. | 0.196 | 0.024 | 0.368 | 0.053 | 0.285 |
| | (*) | H0 | **H1** | H0 | H0 | H0 |
| General rate based on three criteria (age, gender, settlement) | N | 1600 | 1565 | 1551 | 1561 | 1549 |
| | Chi-square | 62.514 | 50.986 | 11.947 | 44.539 | 17.164 |
| | d.f. | 11 | 11 | 11 | 11 | 11 |
| | Sign. | 0.000 | 0.000 | 0.368 | 0.000 | 0.103 |
| | (*) | **H1** | **H1** | H0 | **H1** | H0 |
| | Median | 0.75 | 0.75 | 0.50 | 0.50 | 0.75 |
| | Chi-square | 44.167 [b] | 17.965 [c] | 10.004 [d] | 36.658 [e] | 20.850 [f] |
| | Ass.sign. | 0.000 | 0.082 | 0.530 | 0.000 | 0.035 |
| | (*) | **H1** | H0 | H0 | **H1** | **H1** |

(*) **H1**–Correlation between variables is confirmed. H0–Correlation between variables is not confirmed. Frequencies less than 5 were expected in 0 table cells. [b.] Minimum expected frequency–15.1. [c.] Minimum expected frequency–7.7. [d.] Minimum expected frequency–31.9. [e.] Minimum expected frequency–30.3. [f.] Minimum expected frequency–14.0. Source: authors' research.

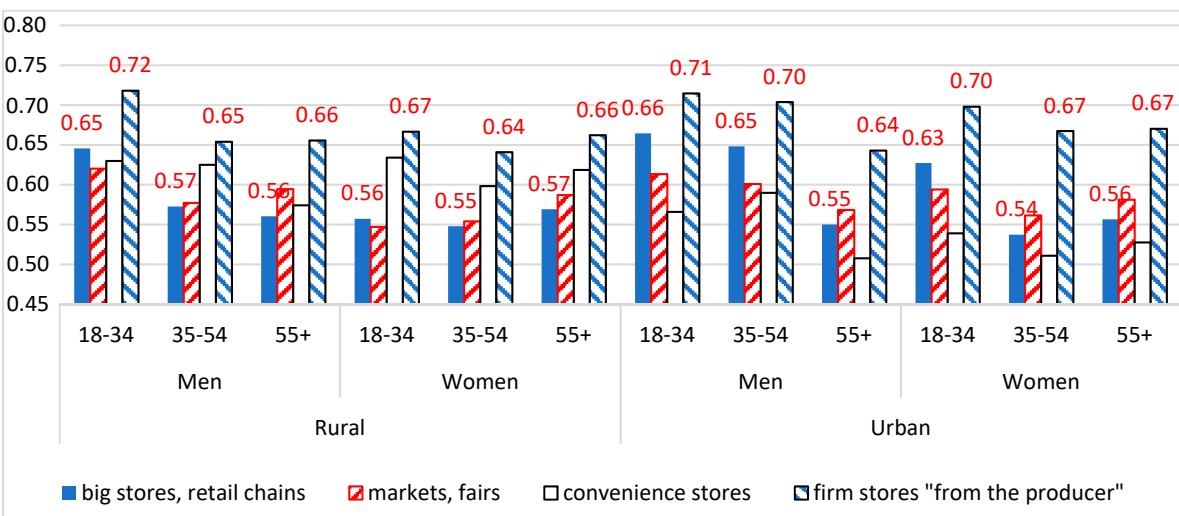

**Figure 2.** Average index of trust in regional sales operators in terms of food quality and safety for settlement type, gender and age. Source: authors' research.

Thus, people most of all trust in quality and safety of food products bought in "firm stores", regardless of their status and income. However, such products are more expensive, and predominantly bought by wealthier citizens. Economic affordability of such products declines simultaneously with income, which explains the relative growth in preference of markets and fairs. Unlike European countries, where products are bought at regular Sunday and holiday fairs, which are highly trusted and in demand, in modern Russia this tradition is lost.

It is indicative that the level of education and the social status do not give statistically significant difference. It was assumed that differences were mainly driven by affordability. The level of formal education has little importance; and cultural characteristics in the design of the study cannot be identified. More research is needed to test Hypothesis 3. It seems to us that the influence of formal education is becoming less and less significant in the formation of a person's level of awareness in everyday life. We received indirect confirmation of these conclusions in our other studies, e.g., [42].

There are two related circumstances to explain why women, on average, trust in quality and safety of food less than men do. First, women in Russia have a significantly lower income level. Second, in the age group over 50, there are reasonably more women than men. These two factors lower the level of trust in food products in general and sales operators in particular. However, age cohorts and settlement types do not erase gender differences in assessments. Probably, women are more likely to care about food quality and safety.

Answers to question Q2 enable us to identify preferred purchasing practices and food affordability.

First of all, the rating of sales operators' market share does not depend on the type of producer. Market shares are specified as follows: the first number reflects the share of purchase place preferences for all products in general, the second number is the share of products from local producers. Convenience stores are extensively favored as the main place for purchases by regional residents (76%/71%). They are followed by hyper-, supermarkets, malls (74%/66%)–usually located far from home and requiring time to reach; food markets (26%/24%); food bases (10%/8%); fairs (9%/11%); directly from the producers (6%/7%); and the Internet (5%/3%).

Secondly, there is no significant gap between convenience stores and retail chains.

Thirdly, chain stores and malls lag behind in the ranking due to weak representation of locally produced food.

Structural features of grocery stores' target audience are mainly influenced by income and age. The older the respondent is, the sooner he would prefer an outlet close to his home/work. The younger the respondent is, the sooner he would go to hyper-, super-markets and malls. The higher the income is, the more likely a person chooses hyper-, supermarkets and malls. The target audience of markets and fairs is practically independent of income, as mostly middle-aged people favor these outlets. The target audience of the food bases is the poorest and older residents of the region. In addition, 7–9% of the population do purchase directly from the producer; the youngest respondents practically do not use this sales channel. Online grocery shopping is common only in "the rich" group (21%). Online practices are not yet widespread in any other social group, fluctuating within the range of 5–6% for urban residents and citizens aged 25–44.

Whereby, according to experts, rural residents have few opportunities to buy the needed amount of food; instead, they involuntarily rely on self-provision:

> *"For a city and a village where, say, 400 people live, this is normal, everything is available. But small villages, remote, with a population of 80 people: there is no store, and nothing is available to them. There is no infrastructure, roads, and it is unlikely to change in the nearest future".*

Experts emphasized that not all outlets can respond to customer requests while organizing their internal work. Chain hypermarkets and supermarkets operate according to the list of goods formed at the head federal or regional office.

> *"Nobody asks whether we want these products (to sell) or not. We trade what we have in the program, what the federal office supplies us".*

> *"Let's suppose a client asks me for cheese produced in the Tyumen region. I cannot order it, because I do not have this product in the program".*

In this case, much depends on chain store directors and their initiative. They prepare an offer, send it to the head office for consideration and, if a decision is positive, change the supply contract. The interaction between retail, producers and local suppliers remains a problematic issue. As an expert from the group of municipal management argues, "*conclusion of several agreements at the level of regional and municipal government allowed businesses in agriculture to reach decent standards while processing, packaging, barcoding, and certification. This gave the green light to sell part of the products through retail chains. At present, statistics show that local products account for 30–40% of the regional food sales*".

Moreover, small producers may not always meet the requirements.

> *"Chains cannot work with businesses which are underdeveloped. The chains have technologies that do not allow working with an artisanal manufacturer".*

> *"All agricultural enterprises are passionate about production processes. They often don't have time to advance their goods".*

> *"Even if I can find time for the development of brand and packaging, so that later try selling to chains, I don't need all this, as I'm not that "big". I think the problems that larger enterprises have with chains are most likely to be the drawbacks of their manual mode of management".*

Not surprisingly, people prefer to buy food in grocery stores and supermarkets; and these findings are fully consistent with the trust rating (Table 1). Analysis of preferences for sales operators allowed for identifying the types of products that people are more likely to purchase daily (or more frequently) in stores near home/work. These are primarily bread and baked products (68%), to some extent, dairy products (49%) and grocery products (46%) (Table 3). Food that people try to buy in hyper- and supermarkets practically do not depend on the product type (except for bread). It is noticeable that in large stores people try to do all the required purchases at once. All other products are fairly balanced between the first two options.

**Table 3.** Answers to question Q2. Where do you usually buy the following groups of products?—percentage of the respondents.

| | Convenience Store (near Home/Work) | Hyper-, Supermarkets | Markets, Fairs | Food Base | Producer | Internet | More than One Answer | No Answer | Total |
|---|---|---|---|---|---|---|---|---|---|
| 1. Bread and baked products | **68** | 20 | 3 | 0.2 | 2 | 0.1 | 5 | 1.7 | 100 |
| 2. Confectionery | **46** | 40 | 4 | 1 | 2 | 0.3 | 4 | 2.7 | 100 |
| 3. Grocery products (flour, cereals, etc.) | 36 | **48** | 7 | 3 | 1 | 0.1 | 3 | 1.9 | 100 |
| 4. Meat and meat products | 22 | **37** | 20 | 4 | 9 | | 3 | 5 | 100 |
| 5. Poultry meat, processed products | 25 | **46** | 15 | 3 | 4 | | 3 | 4 | 100 |
| 6. Sausages, smoked products | 31 | **46** | 10 | 2 | 2 | 1 | 4 | 4 | 100 |
| 7. Fish and seafood | 24 | **44** | 17 | 3 | 3 | 1 | 3 | 5 | 100 |
| 8. Eggs | **39** | 36 | 9 | 2 | 6 | 1 | 4 | 3 | 100 |
| 9. Milk products | **49** | 34 | 5 | 2 | 4 | 0.1 | 4 | 1.9 | 100 |
| 10. Butter, cheeses | 38 | **46** | 7 | 2 | 2 | 0.2 | 4 | 0.8 | 100 |
| 11. Fruits and berries | 27 | **46** | 14 | 3 | 3 | 1 | 3 | 3 | 100 |
| 12. Vegetables, grown in a greenhouse | 25 | **45** | 13 | 3 | 5 | 1 | 3 | 5 | 100 |
| 13. Vegetables, grown in an open ground | 21 | **35** | 12 | 2 | 4 | 1 | 2 | 23 | 100 |
| 14. Alcohol drinks | 33 | **49** | 3 | 1 | 1 | 0.4 | 3 | 9.6 | 100 |
| 15. Non-alcohol drinks | 41 | **45** | 3 | 1 | 0.3 | 0.4 | 4 | 5.3 | 100 |

Modal signs are highlighted in bold (types of sales operators depending on a type of food product).

There is a third group of products, which is usually bought at food markets, fairs and food bases (Table 3). This group includes meat and meat products, poultry and poultry products, sausages (containing almost two times less meat), fish and seafood, fruits, berries, vegetables. In addition, people buy meat (9%), eggs (6%), dairy products and poultry (4%) from the producer (apparently, these are purchases from farmers) a little more often. A very small share of consumers buy products via the Internet (generally less than 1%); and these orders, as a rule, include a wide range of products.

As expected, rural residents are less likely to buy food in supermarkets and other big stores. However, this does not influence the ranks of sales operators. In cities, the third group of products is significantly more often bought at markets and fairs: meat and meat products (24%); poultry meat (17%); sausages, smoked meats (12%); fish and seafood (20%); eggs (10%); fruits and berries (17%); vegetables grown in open ground (19%); and vegetables grown in a greenhouse (16%).

Despite the fact that local producers are obviously more trusted, mostly rural residents buy food directly from them. This practice is uncommon among urban population. Rural residents directly buy meat and meat products (15%); poultry meat (6%); eggs (10%); and vegetables of all kinds (8%) from the producer. Thus, the potential of firm stores, where

local farmers and small enterprises can sell their foods, is obvious, as the majority of citizens notes a significantly higher level of trust in quality and safety of these products.

However, notable differences in consumer practices and trust between residents of rural and urban areas have nothing special. For example, studies by Contzen and Crettas show that, even in the richest countries (for example, Switzerland) many farming families face socio-economic problems. This occurs despite the fact that Switzerland is a wealthy country with efficient instruments of agrarian policy. In eleven out of 32 interviews, respondents indicated that they had to be very careful in managing their food expenses and that self-sufficiency is a way to contain costs [43].

Among other things, differences in consumption practices of the wealthiest citizens, services workers, education and culture turns out to be noticeable, while social status at the main occupation (number of subordinates) does not have a significant impact. Higher education increases the traffic to firm stores "from the producer" by 5%, but these differences are not statistically stable.

Experts from a group of trade organizations and food processing enterprises addressed to modern consumption practices. When choosing a product, the buyer adheres to several purchasing strategies. Products of daily consumption are more often bought in convenience stores, as well as in hyper- and supermarkets, if a person does periodic big purchases with a shopping list. The assortment of chain stores provides a wide selection of all food groups, including products from local producers. Chain stores adapt to the changes in the structure of demand: " . . . *expensive products are gone, the grocery basket has decreased, and local producers have greatly expanded. Many cafes and restaurants do home delivery... We do not sell sushi, pizza, and pies now, because this is no longer sold in the store*".

When choosing a product, the preference is given to processed food (for example, washed vegetables, cut poultry, etc.). Quality is a very important determinant of choice. However, experts define "quality" differently: a natural product without additives harmful to health; a product made of high-quality agricultural raw materials; a fresh product. Experts (representatives of trade and producers) pay special attention to "freshness", emphasizing that the focus on selling fresh goods can be a competitive advantage of firm stores (or outlets that directly interact with producers).

Sales operators note the increased nutrition awareness:

*"Customers have become more demanding. Now they understand all the nuances and rules. They have the opportunity to travel to other cities and countries. And they demand quality today. Not only the price is important, but also quality, packaging, and ... freshness".*

*"Today a person needs a product to be packed, cut. In order not to bother with this fish, but just open the package and be certain that it is fresh." Here is a very important point. Today, joy should come from quality... This is what we need to strive for: we want to ensure that our product could be presented this way".*

Consumer preferences are changing to some extent due to healthy lifestyle and its slogan: "*People now choose food more thoroughly . . . they began to eat more vegetables and fruit, but less meat . . . chicken fillets are consumed more frequently*".

The concept of "healthy food" among Russian consumers remains very narrow. Actually, no one thinks of how products "get" on the shelves, how they are stored, what methods are used in their production, whether these products are safe for health. However, in the world's experience, these factors have long been considered decisive (see, for example, the works of Baum & Pauls [44]; Quintanilla et al. [45]; Hung Anh et al. [46]).

According to experts, the price remains significant for consumers:

*"The consumer is not ready to overpay...".*

*"People walk, look and choose, compare. If a product is being promoted, they ... will buy it".*

*"The choice of the price-quality ratio is quite difficult. A really high-quality product cannot be cheap".*

*"Our consumer is moving towards a falsified, low-quality product. Which is unfortunately sold on the shelves of our main retail chains... Due to this, we surely have a very strong violation of competition... People "vote with their ruble" for cheaper goods, that's why producers of high-quality food suffer".*

The high awareness of modern buyers makes them more demanding not only for products, but also for quality of services provided by a store. Moreover, buyers are ready to actively defend their rights, directly contacting supervisors or regulatory authorities.

*"A considerable number of unscheduled checks takes place upon complaints of clients: ...a number of complaints we have now decreased slightly. It is about 13,000. But last year there were more than 15,000. Imagine, it's for 200 working days...".*

Several experts (representatives of sales operators) emphasized that buyers more and more often express their negative opinion, without objective assessment of the product. Describing the situation, one of the experts even used the term "consumer extremism".

*"An annoyed consumer calls . . . and says: "It is written on your dumplings (pelmeni) that it is a plant product? You said it was meat". Then he hung up. However, the answer is very simple. The plant product is flour".*

*"...Meat has an unpleasant odor, it seems to smell boar taint, but they do not know how a boar really smells".*

Such respondents rate the most important characteristics of food products: price, freshness, expiration date and taste. An absolute majority of buyers share this opinion (about 80% of respondents). The fourth criterion, taste, is important for 60% of respondents. The other five criteria–place of production (37%), presence of GMOs (33%), presence of preservatives (29%), packaging, and brand (17%)–collectively gain 40%.

Freshness and expiration date are almost equally relevant for all social groups. Price, as expected, is less important for "the wealthy" individuals, but is a critical characteristic for small town residents and the elderly (respondents over 65), as well as for young people (under 24) and "the poor". Yet, as the financial situation improves, the criterion of taste becomes visibly more attractive: 47% for "the poor" group and 75% for "the wealthy" group. These findings correspond to our previous studies (see for example [47]).

Product brand as a choice factor, on average, has the least influence, as only 17% of the consumers pay attention to it. However again, with the growth of income, this component becomes more and more decisive, as 26% of "the wealthy" are brand-conscious. Interestingly, the brand is evidently more influential for the young respondents (31%).

In general, about 30% of the respondents pay attention to such characteristics as the "presence of GMOs" and "without preservatives". These are mainly middle-aged buyers, residents of the regional capital, more often women. Note that, according to the conclusions of experts (representatives of producers and sales operators), the information on the package of semi-finished products "without preservatives" is nothing more than a marketing trick, because even sugar and salt are preservatives.

*"No GMO campaign in the media or "no preservatives" text on the package—we understand that this is advertising. And good-faith producers end up suffering losses. Because a truly environmentally-friendly product is the technology, combining growing, production, transportation and storage".*

Online food shopping is very unpopular. According to expert opinion, Internet sales mainly occur in cafes and restaurants; ready-to-eat food delivery has filled this market, practically displacing this service from retail chains and big stores.

Overall, the authors have analyzed Western Siberian consumers' behavior and revealed their perception and awareness of food quality and safety, as well as trust in various product groups and outlets.

Results of the authors' hypotheses verification.

**H1** is confirmed. The majority of the respondents (over 60%) trust in quality and safety of food.

**H2** is confirmed. Regional residents' trust in quality and safety of food is higher than in government, governor, local authorities and business community.

**H3** is confirmed. Firm stores are the most trusted, regardless of socio-economic characteristics. However, the availability of "from the producer" food declines in line with income. Therefore, mainly wealthier citizens buy these products. Big stores (hyper- and supermarkets) and retail chains are rated the second in the list.

**H4** is partially confirmed. The main factors influencing food consumers' behavioral features are income, age, gender and settlement type. The influence of other factors is either not revealed, or partially revealed (in relation to specific groups and sales operators).

**H5** is confirmed. The level of trust in quality and safety of food and the level of trust in sales operators increases in line with income. The qualitative analysis of interviews allows arguing that wealthier respondents pay less attention to price and focus more on such characteristics as taste and brand, while poorer respondents try to economize on red meat, substituting it with cheaper meat processing products. In addition, education does not show a statistically significant difference. It seems that the influence of formal education is becoming less and less significant in the formation of a person's level of awareness in everyday life.

**H6** is not confirmed. Education does not affect the behavioral characteristics of food consumers.

**H7** is not confirmed. There is no difference between urban and rural residents when choosing outlets.

**H8** is confirmed. The younger the respondents are, the higher their trust in firm stores (brand choice) and retail chains (sales regulation choice).

**H9** is not confirmed. There is no conflict of interests between sales operators and consumers. The experts' opinion helps reveal some contradictions between consumers' expectations and retail chains' internal policies. There is a dependency of chain stores on the federal policy, which limits the amount of local food on shelves. However, the share of food produced by local farmers and enterprises, demanded and trusted by consumers, in chain stores is steadily growing due to new agreements.

## 5. Discussion and Conclusions

This paper analyzes consumer food choice in the context of accessibility, affordability, and trust in quality and safety. Based on the materials of a regional study, implementing qualitative and quantitative methods, some trends in the relationships between food buyers and sellers are identified. The obtained data are not exclusively territory-specified, as sales operators act throughout Russia in compliance with unified rules. Therefore, interaction problems are common for the entire country.

In spite of the increase in the overall nutrition awareness, the global policy for maintaining food security has not reached its target–mass consumer. Due to insufficient understanding, the increasingly popular concept of "conscious consumption", that the government tries to advance, contradicts the international organizations' vision.

Slowly but steadily, a "new model" of food consumer behavior is being formed. The credibility of "from the producer" food is higher, but most of the purchases are made in the convenience stores. The wealthiest social groups demonstrate a greater willingness to gradually switch to conscious consumption based on qualitative characteristics rather than quantitative. In general, people rarely buy products for future use, preferring small, regular purchases, and they are also attentive to the choice of stores. Almost all respondents clearly distinguish between types of products and producers, can accurately or almost accurately categorize the purchases by frequency and nomenclature.

The choice of quality food products is becoming an important sign of achieving a certain social status: average and above the average income. Buyers are becoming more demanding, and they are more likely to complain to regulatory bodies. Product packaging remains one of the most important signs that a buyer uses to judge product quality: buyers carefully study the text on the package and product characteristics.

A decreasing consumption of red meat is substituted by poultry meat. There is also a decline in the consumption of fruits and dairy products (this is especially true in the more expensive segments: cheese, butter, yoghurts). On top of that, the impact of global brand awareness on food sales has been greatly exaggerated. According to the respondents, the most significant signs for them are price, freshness, expiration date and taste. Brand influence is taken into account only by 17% of the consumers, but the situation changes with the growth of consumers' well-being, as 26% of the wealthiest respondents pay attention to this characteristic. In addition, the higher rating (above 30%) of such factors as "no GMO" and "no preservatives" is noted, which, as a rule, has only an advertising effect and only slightly reflects the quality and safety of a product.

The development of farmer stores is primarily limited by economic reasons. Such products are more expensive and, in the face of declining purchasing power, individuals are shifting to cheaper options–chain or small convenience stores. It creates significant restrictions on the development of short supply chains and affects the sales of food products from local markets, while the demand for such formats in Russian society is very high (as well as confidence in quality and safety).

The authors claim that the Russian Federation interprets the food security concept in a peculiar way: it not only contradicts the global one, but also contains elements of isolationism. However, such a concept did not come out of nowhere, and it is, to some extent, justified, since it is dictated by attempts to ensure the security of the state. The problem of "connectivity" is topical for Russia, as some large territories are hardly inhabited by people. A part of the population understands that the geopolitical situation is not always favorable for Russia. This is why the idea of import substitution fits well into their picture of the world, predetermining the relatively wide support of the isolationism policy.

However, there is evidence that food import is a less crucial threat than a decrease in the purchasing power: the growth of retail food prices is not balanced by population income changes [48]. With the fall in a real disposable income, Russians began to consume less expensive imported food products (such as vegetables, fruits and berries, meat and meat products), replacing them with cheaper domestic (bakery goods and pasta) [49]. In these conditions, the massive state-budgetary support for agriculture is questionable. As Belova notes, to assess the level of food security, indicators of economic affordability should be used instead of the share of imports in food resources [50].

The policy of import substitution leads to negative consequences for the population. Despite the large size of the territories, there is no problem of accessibility (stores' shelves are not empty); there is no hunger (residents have sufficient money to ensure the needed level of consumption). However, due to a misunderstood concept, the authorities are building barriers for local producers who are now unable to sell their goods quickly.

Generally, people have a higher level of trust in the quality and safety of food produced close to their place of living, but have to buy in chain stores. Therefore, at the declarative level, local producers are supported, but in fact, chain stores located nearby, lure away the customer traffic: a person does not need to go to the market (only if they have not managed to find everything they wanted).

The existing regional grant policy to support local producers came into conflict with the federal one. The Federal Antimonopoly Service of the Russian Federation fined the Tyumen Oblast for "buying Tyumen products", because it might be perceived as a war between regions of the country.

The authors suggest that the authorities can have an impact on changing chain stores practices. Local producers need not only need to be helped to promote their products through fairs and markets (demanded by a tiny part of the population), but also to improve the conditions for selling products to chain stores, which currently buy very limited amounts of local food.

However, it should be noted that, according to experts, meat bought on the marketplace is less safe than the meat bought in a store. The population does not understand this. Representatives of federal retail say that the farmers' products do not meet the require-

ments of quality certification, while marketplaces do not check the chemical composition. Livestock can be fed with waste or grazed on the unsuitable land (for example, near highways). This is alarming, especially taking into consideration that recent literature shows an increasing interest in the consumption of local food [51].

Future research implications. The commonly used grant policy to support agribusiness is questioned. Grants only help dealing with problems of one kind–lack of money to initiate operation, and ignore significant obstacles that small- and middle-sized local producers face entering retail. Hence, primarily wealthier and more educated people with a rather "conscious" approach to consumption are ready to search and purchase the needed local food, evaluating consequent quality risks.

Conditions for improving the quality of local producers' food may be a promising area of future research, because there are many statements in the literature that show that short supply chains have an important role in sustainable rural development, positively impacting the local economy, job creation, income and land use [52].

Further scientific studies on similar problems should focus more broadly on the mythology of some issues, for example, the dangers of GMOs or the importance of bright and visible branded packaging; and vice versa, they should aim to expose the real danger of economizing on quality food, neglecting nutrients balance and modern person's lifestyle needs [53].

It is urgent to understand properly possible success factors of short food supply chains in the region. Obstacles may occur in three stages: chain creation, product development and access to the market. These findings may provide insightful information for local producers, facilitating sales through understanding peculiarities of different channels: online, local communities (small outlets that cover large territory), chain stores, HoReCa (agreements with hotels, restaurants and catering), direct (inviting customers to farms/enterprises) [54].

Practical implications. Research results are useful for enhancing food security policy at the federal and regional levels. Not only firms and professionals in marketing, but also policy-makers and social organizations can adopt the obtained conclusions.

Population's readiness to support short food supply chains is shown, revealing opportunities for local territorial community development. The authors' recommendations focus on physical accessibility and economic affordability of food products from local enterprises, which are demanded by consumers in terms of trust in quality and safety.

There is a contradiction between affordability and accessibility. It can be claimed that the regional food retail turnover is restrained due to insufficient representation of local products on the shelves. It is expected that larger sales will allow local producers to decrease purchase prices, which will lead to higher affordability of their goods, as well as facilitate nutrition balance improvement.

The authors argue that local producers are lacking opportunities to expand their sales network and need support from the authorities. The consistent development of legal interaction between the government and local business is expected to promote more sustainable development of territorial communities and allow people to consume trustworthy and quality food.

In turn, the federal retail calls for a higher level of confidence in the production process and control over quality and safety. Currently, representatives of big chain stores doubt small businesses' operation. Hence, the governmental support policy may be directed towards production regulation, requiring firms to systematically report on a set of indicators that are consistent with demands for quality and safety of modern retail.

**Author Contributions:** The authors equally contributed to the original draft preparation. All authors have read and agreed to the published version of the manuscript.

**Funding:** The reported study was funded by RFBR, project number 20-011-00087.

**Informed Consent Statement:** Informed consent was obtained from all subjects involved in the study.

**Data Availability Statement:** The data presented in this study are available on request from the corresponding author.

**Conflicts of Interest:** The authors declare no conflict of interest.

**Appendix A. Questions Used in Expert Interviews**

1. What industry does your company (organization) operate in?

2. In your opinion, what are the priorities for ensuring food security?

3. What can you say about modern food consumption practices (a question to the experts working in sales operations)?

4. What do you think about consumer protection in terms of food safety and quality?

5. Do local agricultural producers provide a sufficient level of quality?

6. What qualitative changes have occurred in the industry over the past five years?

7. What difficulties and contradictions do you face?

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
