# Peer review of "Consumer Trust in Quality and Safety of Food Products in Western Siberia"

_agronomy, doi:10.3390/agronomy11020257_

Round 1

Reviewer 1 Report

Necessary corrections:

1. The word “elastic” needs to be re-think and re-formulated. “The level of trust is elastic across various social groups” (line 22).

Please take a look at: Cambridge Dictionary <elasticity> (flexibility of functions) is used in economics as a measure of the sensitivity of one variable to changes in another (https://dictionary.cambridge.org/pl/dictionary/english/elasticity. Derived on 31.12.2020)

2. New external and internal challenges, both economic and political such as …. (line 30-32). It will be better to order this important introductory sentence. F. ex. “New external, such as: … and internal challenges, such as …….. increase relevance of countries and regions.

3. The part on: “Despite the cooperation of Russian government with FAO …” (lines: 59-64), is very interesting but not readable to readers from, different than Russia, parts of the world.

It needs to be clarified, f. ex. :”it is understood somewhat differently” (line 60). So how? Explain it. Major indicators (line 63). So which indicators?

4.The hypotheses have been correctly formulated (see line 110-115), but they need to be positively or negatively confirmed (see line 536-554). From the statistical point of view, the hypothesis is <not true>. It is confirmed, or not confirmed.

5. What do authors mean by “<We> implement the FAO’s food” (see line 41).

See also: “<We categorize into …>” (line 138)

I hold the opinion of those scientists who fulfill the obligation to maintain the formal and impersonal mind-setting across scientific discourse.

6. From the methodological point of view, the aim of the paper can never be “an analysis”, because an analysis is the tool. There is a need to re-formulate the aim of the article (line 102).

7. There is no definition, explanation and discussion on <safety>, which is pointed out as the basics of the basics of the paper already in the title. There is a need to provide this section, after 2.2. Trust to food products, and before 2.3. Local producers and food supply chains.

8. If <safety> is replaced with <security>, then it needs to be clarified and standardized in the whole text.

9. Try to omit such meaningless words like: actually, generally, nowadays, current, etc.

What does <current> support programs mean? When is current? (line 212).

What does nowadays mean? Does it bring closer to the matter? (line 122).

What does normally mean? (line 37).

 10. There is no information why those research methods have been chosen (line 261).

11. As the researchers, as the investigators of the matter, try to explan why men and women put different ranks to big stores, retail chains and / or covenience stores (line 316) and later on (line 317).

12. Figure 2 (line 318-320) needs to be re-formulated. What if I am interested in getting the results of the research on female living in the rural being in my early twenties? Can I find it?

13. Change the word from <propaganda> to slogan (line 466) for slogan in order to mainain neutral style (not pejorative, often used in the Soviet times).

14. The phrase <promotional product> does not exist in BrE. It is „a product on promotion“, or „product on offer“. It is a typical language mistake for Slavic mother-tounges.

15. Future research implications (line 627-634) and Practical implications (line 635-640) are of a great value that is why they need to be more developed.

Positives:

  1. The article tackles with a very important and interesting subject.
  2. The structure and “body” of the paper is correct.
  3. The reasoning is conducted logically.
  4. The paper has been written in a communicative language.

Author Response

Dear reviewer,

We are sincerely grateful for your valuable comments. These insightful recommendations allowed the authors to improve this manuscript significantly.

Please, see our answers to your corrections.

Reviewer 2 Report

The paper treats the interesting subject of food quality and safety and the consumer trust. It brings interesting insights in both the workings of the food distribution chain in Russia as well as consumer behavior. The paper has merit and could add value to the field but there are some shortcomings that need to be addressed beforehand.

The authors should consider changing the way the hypotheses are formulated in more quantifiably way. For instance: “H1. People generally trust the quality and safety of the food they buy.”. What does generally mean? What percentage of population would need to trust the food quality to be considered generally? What does it mean that they trust it? A hypothesis formulated something like “People have higher trust in the food safety and quality bought from retailer 1 than from retailer 2.” would be a testable hypothesis (with a Kruskal-Wallis H test). The same thinking applies to the rest of the hypotheses.

Even though the scale that the trust was measured on was coded using decimal values (1, 0.75, 0.5, 0.25, 0), it still remains an ordinal scale (5, 4, 3, 2, 1). This causes a series of issues in the interpretation of the results. Even though there are different schools of thought on the subject, my suggestion is to use the median and interquartile range as indicators instead of the mean and standard deviation. As the distance between measurements is not equal, the mean cannot be interpreted correctly.

The fact that the measured data is on an ordinal scale means that you cannot use parametric methods such as ANOVA. You need to use the non-parametric alternative, the Kruskal-Wallis H test.

When reporting the results of statistical tests (such as ANOVA or in this case Kruskal-Wallis) it is common practice to report them in the APA style. Besides reporting the F value, one should also report the degrees of freedom, p-value etc. I urge the authors to search for how to report their results in APA format.

The values in Table 2 don’t add up to 100%. If there are invalid or missing responses, this should be indicated in another column so that the percentage of responses add up to 100%.

The authors need to pay great attention to the wording when testing hypotheses. The agreed upon formulation of hypotheses is by stating the null hypothesis. The null hypothesis states that there is no significant difference between groups, or there is no significant relationship between variables. By performing statistical tests, the null hypothesis can either be confirmed or not. If it is not confirmed, then the alternative hypothesis is considered (that there is a significant difference or relationship). Please rephrase the results in this way.

The English writing should be improved. There were some spelling errors as well:

95 “is achieved” instead of “in achieved’

149 error in citation “[van22 Rijswijk]”

209 “found” instead of “fined”

456 “Sales” instead of “Seles”

546 “red meat” instead of “read mean”

Author Response

Dear reviewer,

Let us kindly thank you for the insightful recommendations. These important corrections allowed the authors to improve our manuscript substantially.

Please, see our answers to your corrections.

Round 2

Reviewer 2 Report

The article has improved since the first submission. The results are presented more clearly as a result of the new graphical representations. The non-parametric tests are suitable to the data type and the results include statistical significance. Although the "level of trust metric" is still referring to an ordinal variable, I trust that the authors have used the Kruskal-Wallis test appropriately.

The English writing can still be improved. For example, the phrase "level of trust to food" should be "level of trust in food". Another example is in figure 2 where "Man" and "Woman" should be replaced with "Men" and "Women". The authors should ask for the help of a native English speaker or an editing service.

Author Response

Dear reviewer,

The authors are grateful for your expertise and essential advice. Your comments helped improve the article substantially.